Impacts of tropical cyclones on Northwest Atlantic seabirds: insights from a Category 1 hurricane

Burt Tori V. 1 tburt@mun.ca
http://orcid.org/0000-0002-1448-1461 Blackmore Robert J. 2
http://orcid.org/0000-0003-0059-6900 Collins Sydney M. 1 2
d’Entremont Kyle J. N. 2
http://orcid.org/0000-0002-7461-5564 Ward Christopher R. E. 3
Cunningham Joshua 4
http://orcid.org/0000-0003-0047-8023 Richards Cerren 1
http://orcid.org/0009-0004-5939-7951 Le Taro Fiona 1
http://orcid.org/0000-0002-7842-7317 Wilhelm Sabina I. 3
http://orcid.org/0000-0002-0198-4537 Bates Amanda E. 5 6
Avery-Gomm Stephanie 7
Montevecchi William A. 1
1 Departments of Psychology and Biology, Memorial University of Newfoundland and Labrador , St. John’s, Newfoundland and Labrador , Canada
2 Research Grant and Contract Services, Memorial University of Newfoundland and Labrador , St. John’s, Newfoundland and Labrador , Canada
3 Canadian Wildlife Service, Environment and Climate Change Canada , St. John’s, Newfoundland and Labrador , Canada
4 Wildlife Research Division, Science and Technology Branch, Environment and Climate Change Canada , St. John’s, Newfoundland and Labrador , Canada
5 Department of Ocean Sciences, Memorial University of Newfoundland and Labrador , St. John’s, Newfoundland and Labrador , Canada
6 Department of Biology, University of Victoria , Victoria, British Columbia , Canada
7 Wildlife and Landscape Science Directorate, Science and Technology Branch, Environment and Climate Change Canada , Ottawa, Ontario , Canada
Pimm Stuart
Electronic publication date: 2025 Oct 21
Publication date: 2025
Volume: 13
Electronic Location ID: e20157
Received 2025 May 21; Accepted 2025 Sep 9
Copyright: © 2025 Burt et al.
Copyright year: 2025
Copyright holder: Burt et al.
License: This is an open access article distributed under the terms of the Creative Commons Attribution License, which permits unrestricted use, distribution, reproduction and adaptation in any medium and for any purpose provided that it is properly attributed. For attribution, the original author(s), title, publication source (PeerJ) and either DOI or URL of the article must be cited.
License URL: https://creativecommons.org/licenses/by/4.0/

Keywords: Tropical cyclone, Hurricane, Climate change, Northern Gannet, Morus bassanus, Leach’s Storm-Petrel, Hydrobates leucorhous

Funding: Environment and Climate Change Canada GCXE22C307 Natural Sciences and Engineering Research Council of Canada 006872 Funding was supported by Environment and Climate Change Canada to Tori V. Burt (https://www.canada.ca/en/environment-climate-change.html), Environment and Climate Change Canada to William A. Montevecchi (no. GCXE22C307), and the Natural Sciences and Engineering Research Council of Canada by a Discovery Grant to William A. Montevecchi (no. 006872). The decision to publish the manuscript was part of a contract between author Tori V. Burt and Environment and Climate Change Canada.

==============================
Tropical cyclones are annual occurrences in the western North Atlantic Ocean, where many seabird species are vulnerable to the environmental factors associated with extreme weather events. We summarize the history of tropical cyclones in Newfoundland, Canada, which hosts globally significant populations of seabirds. We examine the interactions that historical tropical cyclones have had with breeding seabirds by plotting the temporal association of Category 1 hurricanes with the breeding phenology of colonial seabirds in Newfoundland and identifying which major colonies have fallen within the pathways of these hurricanes. As a case study, we explore how Hurricane Larry (2021) coincided with increased stranding and mortality of Northern Gannets and Leach’s Storm-Petrels. The breeding seasons of Northern Gannets and Leach’s Storm-Petrels overlapped with all Category 1 hurricanes making landfall in Newfoundland from 1851 to 2024, whereby the central pathways of at least one hurricane passed over the six large Leach’s Storm-Petrel colonies and at least one of the Northern Gannet colonies. For Northern Gannets, a notable stranding and mortality event occurred with at least 146 stranded and 130 dead from September 13 to 24, 2021. For Leach’s Storm-Petrels, 19 birds were observed stranded and 16 died from September 10 to 14, 2021, which was higher than strandings and deaths reported during this period in 2020, 2022, 2023, and 2024. As global climate change drives shifts in the timing, frequency, severity, and attributes of tropical cyclones, we raise the concern that the impacts of tropical cyclones on breeding seabirds may worsen and lead to high mortality in some years.

Introduction

Extreme weather events, often defined as events that exceed seasonal norms based on historical data (Environment and Climate Change Canada, 2024), are projected to become more numerous and severe as global climates change (Knutson et al., 2010; Knutson & Tuleya, 2004; Oliver et al., 2019; United States Environmental Protection Agency, 2016). Tropical cyclones, including typhoons and hurricanes, develop over tropical or subtropical waters (National Oceanic and Atmospheric Administration, 2025a) and drive strong winds, intense rainfall, and tidal surges. Tropical cyclones are measured using the Saffir-Simpson Wind Scale in the Atlantic and eastern Pacific, which classifies hurricanes into five categories representing the storms’ severity (one is the lowest and five is the highest; National Oceanic and Atmospheric Administration, 2025a; Saffir, 1973; Simpson, 1974). The environmental effects of tropical cyclones can impact wildlife across broad regions as storms travel from low to high latitudes, impacting environmental conditions spanning the ocean and land.

Species relying on terrestrial and marine habitats are vulnerable to adverse environmental conditions caused by tropical cyclones (Newell et al., 2015; Wiley & Joseph, 1993). Seabirds, which are already declining globally due to threats such as habitat loss, fisheries bycatch, and introduced species (BirdLife International, 2024; Birds Canada & Environment and Climate Change Canada, 2024; Dias et al., 2019; Montevecchi, 2023), may face additional risks from extreme weather events. Most seabirds are K-selected, characterized by low reproductive rates, delayed maturity, obligate social monogamy, and high parental investment from both males and females, with most species producing only one or a few chicks per breeding season and devoting extensive care to ensure their survival (Hamer, Schreiber & Burger, 2001). As such, seabird populations are at higher risk from the effects of individual mortalities due to prolonged generation time (Nur & Sydeman, 1999). Many seabirds breed in dense colonies on coastal islands and outcrops, subjecting their populations to increased risk from ecological effects, such as disease (Avery‐Gomm et al., 2024; Careen et al., 2024) and extreme weather.

Throughout the breeding season, one member of a seabird pair typically remains at the nest to incubate egg(s) or care for chicks, making them susceptible to harm during storms and localized extreme weather. Strong winds can cause injury, displacement, and death of birds both at sea and at exposed nesting sites (Newell et al., 2015; Wiley & Joseph, 1993). Large die-offs of seabirds are often triggered by extreme weather events, with strong winds producing unfavourable foraging conditions and displacing birds both at the colony and at sea by pushing them inland, which often causes them to become stranded (Bugoni, Sander & Costa, 2007; Diamond et al., 2020; Morley et al., 2016), particularly if birds are otherwise weakened (Anker-Nilssen et al., 2017; Avery-Gomm et al., 2016; d’ Entremont et al., 2021; Robinson et al., 2018). Stranded seabirds often have difficulty regaining flight and are subject to dehydration, starvation, prolonged injury, and predation without human intervention (Burt et al., 2024; Rodríguez et al., 2017). Additionally, citizen reports and rescue efforts of seabird strandings are spatially biased to areas where human activity is highest, meaning remote stranding events go underreported and are difficult to assess. Although seabird strandings do not always result in death, strandings are closely linked to mortality due to these associated risks. Also, extreme rain and storm surges can cause the erosion or flooding of suitable nesting habitats, which may lead to the death of a nesting adult and its chick(s) (Newell et al., 2015; Wiley & Joseph, 1993). If severe weather causes either parent to strand and/or abandon the nesting attempt before the young are old enough to fledge, the chick(s) will likely die (Whitehead & Dunphy, 2022).

Seabirds interact with tropical cyclones on land and at sea in the North Atlantic Ocean, where tropical cyclones occur annually (National Oceanic and Atmospheric Administration, 2025b). These storms are expected during the “hurricane season”, typically when ocean waters are warmest and most conducive to hurricane formation and strengthening, ranging from June to November in this region based on historical data (Landsea & Franklin, 2013; Truchelut et al., 2022). External environmental factors such as the steering flow from the subtropical high, and extratropical westerlies as well as internal factors such as the β gyre of the weather system cause mid-latitude tropical cyclones to initially track poleward and strengthen with high sea-surface temperatures (known as baroclinically-enhanced hurricanes; Elsner & Kara, 1999) before eventually weakening and dissipating upon interacting with large land masses, colder sea surface temperatures, or cold fronts. As tropical cyclones move northward in the Northern Hemisphere, the strongest winds are located in the northeastern quadrant, which is the front-right quadrant relative to the storm motion (Elsner & Kara, 1999). Historical tropical cyclone tracks reveal that tropical cyclones in the North Atlantic Ocean tend to recurve back out to sea from the North American coast (Landsea & Franklin, 2013; Elsner & Kara, 1999) impacting the western and central North Atlantic Ocean which is used by many pelagic seabird species as an important breeding, staging, and foraging habitat year-round (Davies et al., 2021; Wakefield et al., 2021). However, the impact of tropical cyclones on North Atlantic seabirds has not been well documented.

The coastal regions of eastern North America host many large, dense seabird colonies. Globally significant populations of numerous species breed in the Northwest Atlantic throughout the province of Newfoundland and Labrador, Canada, on coastal islands and outcrops (Ainley et al., 2020; d’Entremont et al., 2022; Hatch, Robertson & Baird, 2020; Lowther et al., 2020; Wilhelm et al., 2020; Zabala Belenguer, 2023). At colonies on or near the island of Newfoundland (distinguished from the province of Newfoundland and Labrador, hereafter “Newfoundland”), most colonial seabird species depart from terrestrial nesting sites before late August. However, species that fledge young later in the year, such as Northern Gannets (Morus bassanus; Mowbray, 2020) and Leach’s Storm-Petrels (Hydrobates leucorhous; Pollet et al., 2021), remain at their colonies well into October or later. The seasonal timing of nesting and fledging may translate to differences among species in their exposure to severe weather conditions, particularly tropical cyclones.

In this study, we summarize the history of tropical cyclones in Newfoundland and examine how the timing of tropical cyclones in Newfoundland interacts with the breeding phenology of five seabird species. We also assess which colonies may have been at risk during historical tropical cyclones in Newfoundland for Northern Gannets and Leach’s Storm-Petrels. As a case study, we present the impacts of a Category 1 hurricane, Hurricane Larry, that made landfall on the southeastern shore of Newfoundland and travelled over the island in September of 2021. We synthesize data in the days before, during, and after Hurricane Larry on mortality and on-land stranding of Northern Gannets and Leach’s Storm-Petrels in eastern Newfoundland using both citizen science and structured survey data to document impacts in two Ecological Reserves (Cape St. Mary’s and Witless Bay), acknowledging the inherent spatial bias of citizen reports as a measure of mortality. Finally, we discuss the significance of the impacts of hurricanes on these two species and comment on how changes in the timing and severity of tropical cyclones due to climate change could impact seabirds.

Materials and Methods

Study area

This study focuses on Newfoundland, on the east coast of Canada. Spatially, we define Newfoundland by the land boundary and surrounding islands from the world map data provided in the R “maps” package (Becker & Wilks, 2024). Newfoundland is uniquely positioned at the convergence of multiple atmospheric and oceanic systems. It is in the region of the polar jet stream while also experiencing weather patterns from the mid-Atlantic influenced by the North Atlantic oscillation. Additionally, it lies in the mixing zone between the cold Labrador Current from the north and the warm Gulf Stream from the south. As a result, the southern edge of Newfoundland is subject to direct impacts from the northbound trajectory of tropical cyclones.

History of tropical cyclones in Newfoundland

Newfoundland has a long history of tropical cyclones, including the deadliest hurricane in recorded North American history (1775 Newfoundland Hurricane; Ruffman, 1996); however, there is limited meteorological information on the paths of tropical cyclones available before 1851. Here, we assess the spatiotemporal impacts of tropical cyclones on seabirds breeding in Newfoundland. We examined data from the National Oceanic and Atmospheric Administration (NOAA) (HURDAT2; GIS Map Viewer; Tropical Cyclones 2024; National Oceanic and Atmospheric Administration, 2025a, 2025b) to collate a list of tropical cyclones that made direct contact with Newfoundland from 1851 to 2024 inclusive.

Breeding phenology

The breeding phenology of seabirds determines how and when each species might interact with tropical cyclones. Therefore, we overlaid the breeding seasons of several seabirds that breed in Newfoundland with all Category 1 hurricanes that made landfall in Newfoundland (see above). Gross estimates for the timing of the breeding season were derived from Birds of the World (Ainley et al., 2020; Lavers, Hipfner & Chapdelaine, 2020; Lowther et al., 2020; Mowbray, 2020; Pollet et al., 2021) and external species-specific references (Cairns, Montevecchi & Threlfall, 1989; Kirkham & Montevecchi, 1982; Mahoney, 1979; Nettleship, 1970; Runnells, Montevecchi & Davoren, 2024; Wilhelm, 2017; Wilhelm et al., 2015; Zabala Belenguer, 2023). Based on the assumption that K-selected species are less likely to recover following a large-scale population mortality, only heavily K-selected seabirds (clutch size = 1) that breed in dense colonies in Newfoundland were included in our assessment. Species with >1,000 known breeding pairs in Newfoundland were determined most at-risk in this study, which includes Atlantic Puffin (Fratercula arctica), Common Murre (Uria aalge), Leach’s Storm-Petrel, Northern Gannet, and Razorbill (Alca torda). Notable Newfoundland-breeding seabirds excluded from our assessment are: Thick-billed Murre (Uria lomvia), Northern Fulmar (Fulmarus glacialis), and Manx Shearwater (Puffinus puffinus) due to their low breeding abundance in the study area; seaducks and gulls (including Black-legged Kittiwakes Rissa tridactyla) with clutch sizes >1; and Black Guillemots (Cepphus grylle) which do not breed in dense colonies in Newfoundland. As Northern Gannets and Leach’s Storm-Petrels are the species which remain at their colony sites the latest into the year (Mowbray, 2020; Pollet et al., 2021), we further assess the risks posed by tropical cyclones to their colonies.

Colony risk assessment: Northern Gannets and Leach’s Storm-Petrels

Three of the six Northern Gannet colonies in North America are in Newfoundland (Cape St. Mary’s, Baccalieu Island, Funk Island, Table S1). In the last 7 years, these colonies have been occupied by ~16,500 (counted in 2023; Sceviour, Ward & Wilhelm, 2024) to ~30,000 breeding pairs (counted in 2018; d’Entremont et al., 2022). Northern Gannets at their three colonies in Newfoundland nest on varied slopes, including along steep exposed cliffs and on long, broad flat rock faces, and prefer nest sites that face into prevailing winds. At the species’ southernmost colony in the world, Cape St. Mary’s Ecological Reserve, gannets nest on an isolated sea stack and the mainland (d’Entremont et al., 2022; Montevecchi & Wells, 1984). Newfoundland hosts approximately 50 Leach’s Storm-Petrel colonies, with six large colonies that support more than 10,000 breeding pairs (COSEWIC, 2020), which are Baccalieu Island, Corbin Island, Green Island (Fortune Bay), Great Island (Witless Bay), Gull Island (Witless Bay), and Middle Lawn Island. Baccalieu Island is the world’s largest colony, hosting ~1,950,000 breeding pairs of Leach’s Storm-Petrels (Wilhelm et al., 2020). Leach’s Storm-Petrels at their coastal island colonies dig burrows in the soil or locate rock crevices to use as nesting habitats (Pollet et al., 2021).

To assess the historical risk of tropical cyclones in Newfoundland for Northern Gannets and Leach’s Storm-Petrel colonies, we mapped colony locations against the “best track” of all Category 1 hurricanes that made landfall in the region. The “best track”, as defined by the NOAA, is a smoothed representation of a tropical cyclone’s strength and position during its lifetime, based on all available post-storm data (National Oceanic and Atmospheric Administration, 2025a).

Case study: Hurricane Larry

Hurricane Larry was a Category 1 hurricane (sustained winds of 119–153 km/h) that travelled over the southeastern portion of Newfoundland on September 11, 2021 (Brown, 2021). During this time, Northern Gannets and Leach’s Storm-Petrels were being monitored in Newfoundland and were identified as species potentially impacted by this hurricane. Information on the impacts of hurricanes on seabirds in Newfoundland is scarce, but because Hurricane Larry occurred while impacted populations were already being monitored, it was a clear choice to use as a case study. To understand how the trajectory and physical characteristics interacted with prominent Northern Gannet and Leach’s Storm-Petrel colonies in Newfoundland, the wind swath and best track for Hurricane Larry were plotted over the three Northern Gannet colonies and the six large Leach’s Storm-Petrel colonies (Table S1). We used colony population estimates from 2021 for Northern Gannets and Leach’s Storm-Petrels. To discern the impact of this hurricane on the individual level, we collated observations of stranded and dead Northern Gannets and Leach’s Storm-Petrels in Newfoundland before, during, and after Hurricane Larry. For this study, we considered any seabird that was alive and on land outside of a breeding colony (e.g., on a beach or inland) as “stranded” and any seabird carcasses found as “dead”. We define a “notable” stranding and mortality event as a cluster of ≥100 individual strandings and/or deaths of a single species within 4 weeks in a single region (i.e., province; (Cormier et al., 2024)). Northern Gannets were collected under Canadian Wildlife Service scientific permit no. SC4063 and Leach’s Storm-Petrels were collected under a CPAWS-NL Canadian Wildlife Service permit no. LS2688.

Northern Gannets

Events where Northern Gannets are reported as stranded or dead on-land are very rare: there are few records of such notable occurrences in eastern Canada (Avery‐Gomm et al., 2024; McPhail et al., 2024). Reports from the public in the days immediately following Hurricane Larry indicated that a stranding event had occurred, with observations of injured and stranded live Northern Gannets found on beaches (eBird, 2025; Ward, 2021) in southeastern Newfoundland. To assess the magnitude of strandings and deaths from this event and its association with Hurricane Larry, we collated observations and information gathered from Environment and Climate Change Canada and Cape St. Mary’s Nature Interpreters, who provided observations from the colony and performed rescue efforts and beached bird surveys near the Cape St. Mary’s colony on the days following the hurricane. There were no public reports of stranded or dead Northern Gannets elsewhere on the island of Newfoundland, so our efforts to quantify stranding and deaths were focused solely on the southeastern region of Newfoundland, with rescue and rehabilitation efforts therein.

Researchers arriving at Cape St. Mary’s 2 days after landfall on September 13 focused on rescuing living birds and returning them to the water, although birds stranded on beaches were left untouched. The following day (September 14, 2021), a second assessment of Cape St. Mary’s and the surrounding areas was conducted. Live birds found inland were captured by researchers and returned to the ocean from the nearest location (e.g., released from a beach or a cliff edge). Six injured gannets were taken to a rehabilitation center (Rock Wildlife Rescue), where they later died. Following these two initial assessments and rescue efforts, nine beached bird surveys were conducted. These surveys involved walking along beaches near Cape St. Mary’s, focusing on identifying and counting seabird carcasses and any stranded seabirds. Stranded and dead gannets were identified as adults (>1 year) or subadults (grey plumage). When carcasses were left on the beach, they were placed high above the high tide and marked by observers to avoid double-counting on subsequent surveys.

To better understand the spatial distribution and potential impact of Hurricane Larry on Northern Gannet strandings and deaths, we map the highest recorded number of strandings and deaths from each observed location between September 13 and September 24, 2021, overlaid with the wind swath and best track of the hurricane. We also estimate the total number of strandings and deaths during this period. Since some individuals may have remained at or died at the release location, potentially leading to double-counting, we provide both minimum and maximum estimates of stranded and dead individuals. The minimum count represents the highest single-day count of stranded and dead gannets at each location during the study period, while the maximum count sums all stranded and dead gannet reports across all locations, regardless of potential overlap.

Leach’s Storm-Petrels

Although Leach’s Storm-Petrels are commonly reported stranded in Newfoundland during the fledging period (late September through October; Burt et al., 2024; Wilhelm et al., 2021), an increase in strandings was reported by the public from September 10 to 14, 2021 nearly a month before the mean fledge date for Leach’s Storm-Petrels on Gull Island (October 10; Collins et al., 2023). To assess the magnitude of this stranding and mortality event and its association with Hurricane Larry, we collated observations reported by the public to the Canadian Parks and Wilderness Society Newfoundland and Labrador Chapter (CPAWS-NL, a non-profit charitable conservation group) and the Rock Wildlife Rescue from September 10 to 14 2021. This period represents the start of the hurricane, when the center of Hurricane Larry approached Newfoundland (Brown, 2021) with the potential to affect Leach’s Storm-Petrels foraging offshore, to 2 days after the passing of the hurricane, which allowed for stranded and dead Leach’s Storm-Petrels to be located by the public and reported to rescue groups. Any individuals taken into rehabilitation by rescue groups were returned to the ocean immediately when deemed appropriate and feasible.

Additionally, we included stranding data from a structured daily monitoring effort at a Leach’s Storm-Petrel stranding hotspot, the Quinlan Brothers Ltd. seafood processing plant in Bay de Verde, Newfoundland, Canada (48.0832, −52.8980), located seven kilometres southwest of Baccalieu Island. Daily monitoring for stranded Leach’s Storm-Petrels at the plant has been ongoing since 2020, and the methods for data collection follow those of the “daily morning surveys” in Burt et al. (2024), where systematic sweeps for stranded and dead birds on the property grounds are conducted each day, and stranded individuals are brought to rehabilitation or released back to the ocean as soon as possible. The nature of these data sources does not easily allow for the aging of Leach’s Storm-Petrels found stranded or dead. Most Leach’s Storm-Petrels observations from September 10 to 14, 2021 occurred on September 11, which is just 1 day after the earliest recorded fledging date for this species in Newfoundland (Pollet et al., 2021), and the peak stranding date at the Quinlan Brothers Ltd. seafood processing plant in Bay de Verde in 2021 was on September 25 (Burt et al., 2024). Therefore, it is likely that most, if not all, Leach’s Storm-Petrels reported stranded in Newfoundland from September 10 to 14, 2021, were adult birds.

To better understand the spatial distribution and potential impact of Hurricane Larry on Leach’s Storm-Petrel strandings and deaths, we mapped the highest recorded number of strandings and deaths from each observed location between September 10 and 14, 2021, overlaid with the wind swath and best track of the hurricane. We also present estimates of strandings and deaths during this period. Birds were not double-counted because live birds reported by the public were collected and brought to the Rock Wildlife Rescue or the Canadian Wildlife Service.

Given that strandings and mortalities of Leach’s Storm-Petrel are not unusual in late September through October, we also collated data from the same sources between September 10 and 14 in 2020, 2022, 2023 and 2024 to compare the number of strandings and deaths reported during Hurricane Larry relative to those typically reported in early September.

Results

History of tropical cyclones in Newfoundland

Generally, tropical cyclones that originate in the mid-Atlantic tropical zone and track toward Newfoundland first make landfall along the southern portions of the island (Table 1) as they move northward along the continental seaboard. All tropical cyclones that made landfall with Newfoundland have tracked through during the “hurricane season” from June to November (National Oceanic and Atmospheric Administration, 2025a). However, most tropical cyclones made landfall between late August and October (25 of 28 tropical cyclones, or 89.3%; Table 1), which we define as the peak hurricane season for Newfoundland. The median landfall date for tropical cyclones in Newfoundland is September 11 (mean = September 9; Table 1).

Table 1 Summary of historical tropical cyclone occurrence on the island of Newfoundland from 1851 to 2024 (Landsea & Franklin, 2013).

Name of tropical cyclone	Classification at time of Newfoundland landfall	Date of landfall	Year of landfall	Approximate landfall location	Newfoundland region(s) affected	
Unnamed	Tropical storm	August 27	1851	St. Jacques-Coomb’s Cove	Southern Coast	
Unnamed	Category 1 hurricane	September 23	1866	Cape La Hune	Southern Coast	
1873 Nova Scotia Hurricane	Category 1 hurricane	August 26	1873	Point Lance	Avalon Peninsula	
Unnamed	Tropical storm	August 7	1874	Hermitage-Sandyville	Southern Coast	
Unnamed	Tropical storm	August 20	1879	Channel-Port aux Basques	Southern Coast	
Unnamed	Tropical storm	September 10	1880	St. Lawrence	Southern Coast	
Unnamed	Tropical storm	September 2	1884	Merasheen Island	Avalon Peninsula	
Unnamed	Category 1 hurricane	August 24	1886	Point May	Burin, Avalon Peninsula	
Unnamed	Category 1 hurricane	September 8	1891	Channel–Port aux Basques	Western Coast	
Unnamed	Tropical storm	October 20	1891	Coppett	Southern Coast	
Unnamed	Category 1 hurricane	August 18	1893	Southern Harbour	Burin, Avalon Peninsula	
Unnamed	Tropical storm	October 7	1893	Fortune	Southern Coast	
1898 Georgia hurricane	Tropical depression	October 6	1898	Connaigre Peninsula	Burin, Avalon Peninsula	
Unnamed	Category 1 hurricane	September 15	1899	Cape Race	Avalon Peninsula, entire island	
Unnamed	Tropical storm	September 18	1908	South East Bight	Burin, Avalon Peninsula	
Unnamed	Tropical storm	September 9	1943	McCallum	Southern Coast	
Hurricane Alice	Tropical storm	June 6	1973	Channel-Port aux Basques	Western Coast	
Hurricane Evelyn	Category 1 hurricane	October 15	1977	Channel-Port aux Basques	Western Coast	
Subtropical Storm One	Subtropical storm	October 25	1979	Rose Blanche-Harbour le Cou	Western Coast	
Hurricane Dean	Tropical storm	August 8	1989	Point May	Burin Peninsula, entire island	
Hurricane Luis	Category 1 hurricane	September 11	1995	Patrick’s Cove	Avalon Peninsula	
Hurricane Gustav	Category 1 hurricane	September 12	2002	Burnt Islands	Southern Coast	
Hurricane Bill	Tropical storm	August 24	2009	Point Enragée	Avalon Peninsula, entire island	
Hurricane Igor	Category 1 hurricane	September 21	2010	Cape Race	Avalon Peninsula, entire island	
Hurricane Maria	Tropical storm	September 16	2011	Cape St. Mary’s	Avalon Peninsula	
Hurricane Larry	Category 1 hurricane	September 11	2021	Great Bona Cove	Avalon Peninsula	
Hurricane Fiona	Tropical storm	September 24	2022	Channel-Port aux Basques	Southern Coast	
Hurricane Lee	Tropical storm	September 18	2023	Port-aux-Choix	Western Coast	

Breeding phenology

The breeding seasons of Leach’s Storm-Petrels and Northern Gannets overlapped with all 11 of the Category 1 hurricanes that made landfall with Newfoundland (Fig. 1). Atlantic Puffins have the third longest and latest breeding season, followed by Razorbills; both species overlapped with 27.3% (3/11) of the Category 1 hurricanes. The Common Murre breeding season did not overlap with any of the Category 1 hurricanes (Fig. 1).

Figure 1 The timing of breeding stages of colonial seabirds on the island of Newfoundland, Canada, plotted with all Category 1 hurricanes that have made landfall on the island of Newfoundland from 1851 to 2024 and the years they occurred.

Colony risk assessment: Northern Gannets and Leach’s Storm-Petrels

The central path of at least one hurricane, historically, has passed over each of the six large Leach’s Storm-Petrel colonies in Newfoundland (Fig. 2). Baccalieu Island, the largest Leach’s Storm-Petrel colony in the world, has likely been impacted by three known Category 1 hurricanes (1893, 1899, and 1995), historically (Fig. 2). Baccalieu Island is the only Northern Gannet colony in Newfoundland that fell directly within the path of a Category 1 hurricane. However, cyclones such as the 1873 Nova Scotia Hurricane (Table 1, Fig. 2) and 2011 Tropical Storm Maria (Table 1) have made landfall near or at Cape St. Mary’s. No hurricane has historically passed directly over Funk Island. Three of the eleven Category 1 hurricanes that have impacted Newfoundland between 1851 and 2024 have made landfall and tracked along the western portion of the island of Newfoundland, where there are no Northern Gannet or large Leach’s Storm-Petrel colonies (Fig. 2).

Figure 2 Best tracks of all Category 1 hurricanes that made landfall on the island of Newfoundland from 1851 to 2024 overlaid with locations of large (>10,000 pairs) Leach’s Storm-Petrel (squares) and all Northern Gannet colonies (circles).

Baccalieu Island is both a Leach’s Storm-Petrel and Northern Gannet colony.

Case study: Hurricane Larry

Hurricane Larry passed through Placentia Bay and eventually made landfall near Great Bona Cove on September 11, 2021, at 03:30 UTC (47.3870, −54.5324; Fig. 3; Brown, 2021). The strongest winds recorded in Newfoundland (gusts up to 181 km/h; Brown, 2021) were observed at Cape St. Mary’s Ecological Reserve on September 11, 2021, at 03:00 UTC.

Figure 3 Hurricane Larry wind swath and best track overlaid with locations of (A) large (>10,000 pairs) Leach’s Storm-Petrel and (B) all Northern Gannet colonies on the island of Newfoundland.

The light green area indicates tropical storm force wind speeds (63-93 km/h), and the area of darker blue indicates hurricane force wind speeds (>93 km/h).

Baccalieu Island, located at the northwestern tip of the Avalon Peninsula, also fell in Hurricane Larry’s path and hosts both a Northern Gannet colony and the world’s largest Leach’s Storm-Petrel colony (Table S1). All large Leach’s Storm-Petrel colonies in Newfoundland were within the wind swath of Hurricane Larry, notably including the second and third most populated colonies found in the Witless Bay Ecological Reserve on the east coast of the Avalon Peninsula (Gull Island and Great Island; Table S1). At sea near Witless Bay during Hurricane Larry, wind gusts were recorded at 141 km/h (weather buoy AZMP-STA27, 47.5380, −52.5880, September 11, 2021, 05:00 UTC; Brown, 2021).

Northern Gannets

The total minimum and maximum counts of stranded adult Northern Gannets were 101 and 169 individuals, respectively. The total minimum and maximum number of stranded subadult Northern Gannets was 45 and 67 individuals, respectively. Carcass counts revealed that 101 adults and 29 subadults perished. The majority of gannet strandings and deaths were reported at Point Lance, Newfoundland (46.8091, −54.0780; Table 2), which is 8.1 km east of Cape St. Mary’s (Fig. 4). Cape St. Mary’s is the most likely source colony from which impacted birds were displaced, estimated at 14,600 breeding pairs at the time of this study (d’Entremont et al., 2022), however, no colony abandonment was reported by Cape St. Mary’s interpreters. Cape St. Mary’s was in the path of the front-right quadrant of Hurricane Larry and although the Baccalieu Island and Funk Island colonies were both within the pathway of hurricane force wind speeds (Fig. 3), no strandings or deaths were reported near these colonies (Fig. 4).

Table 2 Surveys conducted by Environment and Climate Change Canada and Cape St. Mary’s Nature Interpreters targeting deceased Northern Gannets at locations near Cape St. Mary’s Ecological Reserve following Hurricane Larry.

Date	Location	Number of stranded adults	Number of dead adults	Number of stranded subadults	Number of dead subadults	
September 13	Cape St. Mary’s	32	0	0	0	
September 14	Cape St. Mary’s	11	22	0	0	
September 14	Point Lance	32	0	11	0	
September 17	Point Lance	27	0	24	0	
September 17	Redlands, Point Lance	67	0	32	0	
September 20	Red Head Beach	0	0	0	0	
September 20	Point Lance	0	30	0	10	
September 21	Portugal Cove South	0	0	0	0	
September 21	Biscay Bay	0	0	0	0	
September 21	Trepassey	0	1	0	0	
September 21	Point La Haye	0	2	0	0	
September 24	Golden Bay	0	46	0	19	

Figure 4 Distribution of Northern Gannet strandings and deaths reported from September 13 to 24, 2021, on the Avalon Peninsula, Newfoundland, plotted against the Hurricane Larry wind swath and best track.

Wind gusts greater than 93 km/hr covered the Avalon Peninsula at this time, and the best track is only shown on the inset map.

Leach’s Storm-Petrels

From September 10 to 14, 2021, 19 stranded and 16 dead Leach’s Storm-Petrels were reported. All stranded and dead Leach’s Storm-Petrels were reported near Bay de Verde and across the northeastern part of the Avalon Peninsula (Fig. 5), locations that are near several important Leach’s Storm-Petrel colonies and intersect the front-right quadrant and highest wind speeds of Hurricane Larry. On Gull Island, Witless Bay Ecological Reserve, the strongest winds recorded in September 2021 were during Hurricane Larry, on September 11 (Richards et al., 2024), the same day that wind gusts at sea reached 141 km/h at a nearby weather buoy (AZMP-STA27, 47.5380, −52.5880, September 11, 2021 05:00; Brown, 2021). As a result, many trees were downed near Leach’s Storm-Petrel breeding sites (R Blackmore and S Collins, 2025, personal observations). In some instances, emaciated storm-petrel chicks were found in uprooted burrows, suggesting that parents whose burrows had sustained damage from the hurricane abandoned their chicks.

Figure 5 Distribution of Leach’s Storm-Petrel strandings and deaths from September 10 to 14, 2021, on the east coast of the island of Newfoundland plotted against the Hurricane Larry wind swath and the best track.

Reports of stranded and dead Leach’s Storm-Petrels between September 10 to 14 were relatively high in 2021, with 35 individuals reported, compared to a total of three birds in subsequent years (Fig. 6). No tropical cyclones made landfall on Newfoundland between September 10 to 14 in 2022 to 2024 (Table 1).

Figure 6 The number of Leach’s Storm-Petrel strandings reported by the public to CPAWS-NL and the Rock Wildlife Rescue, and those found through systematic searches at the Quinlan Brothers Ltd. seafood processing plant in Bay de Verde from September 10 to 14, 2020–2024.

Discussion

The risks posed by tropical cyclones to seabirds in the North Atlantic are poorly understood, and the associated impacts and interactions have not been well documented. In this study, we summarize historical tropical cyclones in Newfoundland and explore how the breeding phenology of seabirds and colony locations are associated with risk. We found that birds breeding at colonies later in the year, including Northern Gannets, Leach’s Storm-Petrels, Atlantic Puffins, and Razorbills, are exposed to higher risks associated with tropical cyclones. We further use Hurricane Larry in 2021 as a case study to qualify the impacts that a Category 1 hurricane had on Northern Gannets and Leach’s Storm-Petrels. We found that Hurricane Larry correlated with strandings and deaths of two species in the North Atlantic.

The path and timing of Hurricane Larry was similar to that of several past tropical cyclones making landfall in Newfoundland (e.g., Unnamed Nova Scotia Hurricane 1873, Hurricane Luis 1995, Tropical Storm Bill 2009, Hurricane Igor 2010, Tropical Storm Maria 2011; Table 1, Fig. 2). Compared to other colonial seabirds with significant populations in Newfoundland, Northern Gannets and Leach’s Storm-Petrels were the only two species whose breeding seasons overlapped with the timing of all past Category 1 hurricanes (Fig. 1). Of the three Northern Gannet colonies in Newfoundland, Baccalieu Island may be the only colony that has seen a direct impact from the paths of Category 1 hurricanes, yet all may have been affected by tropical cyclones historically (Fig. 2). All of the six largest storm-petrel colonies in Newfoundland have likely been directly impacted by Category 1 hurricanes historically, with many colonies overlapping with the location where the tropical cyclones first made landfall (Fig. 2). The wind swath of Hurricane Larry overlapped with all three Northern Gannet and all six large Leach’s Storm-Petrel breeding sites in Newfoundland (Fig. 3).

In 2021, Hurricane Larry coincided with higher-than-expected mortality and displacement (i.e., strandings) of Northern Gannets and Leach’s Storm-Petrels in early to mid-September than are typically seen in coastal Newfoundland (Fig. 6). The stranding and mortality of Northern Gannets and Leach’s Storm-Petrels during Hurricane Larry is concerning, though only Northern Gannets experienced a notable stranding and mortality event. Unlike Leach’s Storm-Petrels that strand on land in association with artificial light at night and during periods of strong onshore winds (Burt et al., 2024; d’ Entremont et al., 2021; Wilhelm et al., 2021), Northern Gannets tend not to strand on land. However, their exposed nest sites on the colony make them exceedingly vulnerable to the effects of the strong winds accompanying tropical cyclones. The nests of burrowing seabirds like Leach’s Storm-Petrels provide refugia from hurricane-force winds, acting as a physical and thermal barrier (Richards et al., 2024), though this can render their nest vulnerable to flooding and erosion in cases of severe rainfall (Newell et al., 2015; Wiley & Joseph, 1993). As such, differences in nest type could have accounted for differences in the magnitude of the stranding and mortality events of Northern Gannets and Leach’s Storm-Petrels during Hurricane Larry. The number of stranded and dead Leach’s Storm-Petrels during Hurricane Larry was lower when compared to the number of strandings typically recorded during the fledging period (Burt et al., 2024), however, the hurricane occurred well before the mean fledge date for storm-petrels in Newfoundland (October 10; Collins et al., 2023), meaning the birds that stranded and died during this event were likely adults. Counts of stranded and dead Leach’s Storm-Petrels were also higher in 2021 compared to reports from the same period in previous and subsequent years (Fig. 6), but these comparisons are limited to the 4-day period before and after Hurricane Larry only and are not representative of strandings throughout the breeding season. Though the strandings documented here appear to be low-impact, markedly high on-land stranding events during the early breeding season present yet another challenge to seabirds who face a multitude of mortality risks associated with climate change.

Stranding and mortality events of this nature are difficult to document as they often have quick onset and widespread effects. Annual monitoring of changes in strandings and deaths is vital to understanding how environmental factors correlate with risk; in the case of Leach’s Storm-Petrels in Newfoundland, citizen science efforts allow for some assessment of inter-annual variability in strandings but do not account for anthropogenic and environmental factors such as artificial light, moon illumination, and prevailing winds (Burt et al., 2024; Wilhelm et al., 2021). Citizen science portals are also valuable for collecting data across a wide geographic range; however, they are inherently biased toward observations from areas with higher human populations and where survey efforts are concentrated (Fig. 5), often with easy access, such as the Avalon Peninsula of Newfoundland in this study. Therefore, it is important to recognize that data from citizen science portals and public reports to organizations, such as those used to report strandings and deaths of Leach’s Storm-Petrels in this study, are not associated with a measure of search effort and most likely underestimate the frequency and magnitude of stranding events. Outside of citizen science, there are often few resources devoted to stranding and mortality events associated with tropical cyclones, especially those that appear negligible to seabird populations. Due to these limitations, information regarding missed detection of carcasses, carcass persistence, and at-sea carcass dispersal was unavailable, likely underestimating mortality. Stranded birds that are rescued are also not guaranteed to survive, further underestimating the mortality associated with these events. Moving forward, beached bird surveys should be conducted near colonies, and carcass counts should be included in a model that incorporates carcass dispersal at sea, missed detections, and carcass persistence to get an estimate of total mortality during stranding and mortality events (Jones et al., 2024; Lavers et al., 2024).

The risks from tropical cyclones for seabirds extend beyond the colony, with impacts on birds at sea. Birds may become injured by high winds or waves while foraging (Wiley & Joseph, 1993) in turbulent conditions, making it difficult to capture prey (Clairbaux et al., 2021; Hass, Hyman & Semmens, 2012). There have been many reports of birds caught in the eye of hurricanes and being transported far from their breeding grounds (Weimerskirch & Prudor, 2019). These individuals often die in the eye of the hurricane or during the journey back to their breeding grounds due to exhaustion and/or injury (Weimerskirch & Prudor, 2019). In contrast, some seabirds seek out tropical cyclones as productive foraging opportunities (Ventura et al., 2024) and fly into the eye of the storm, potentially to avoid strong onshore winds along the outer edge of the cyclone (Lempidakis et al., 2022).

Globally, tropical cyclones can have severe impacts on seabird populations. On Bedout Island, Australia, Cyclone Ilsa in 2023 led to the death of 80% to 90% of booby (Sula sp.) and Lesser Frigatebird (Fregata ariel) populations on the island, with an estimate of ~15,000 booby carcasses observed (Lavers et al., 2024). Similarly, over 350 birds died in southern Brazil following Hurricane Catarina (Bugoni, Sander & Costa, 2007). Though Hurricane Larry did not have population-level impacts via colony abandonment or chick mortality, adult mortality is problematic for both storm-petrels and gannets because of their slow life histories (Hamer, Schreiber & Burger, 2001; Kirkham & Montevecchi, 1982; Mowbray, 2020; Nur & Sydeman, 1999), and tropical cyclones can cause parent abandonment and the loss of nests and eggs (Hennicke & Flachsbarth, 2009). The death of a breeding adult will likely result in the death of the chick and the severing of a pair bond in socially monogamous species (Whitehead & Dunphy, 2022), with the potential for cascading effects.

The risk that tropical cyclones in Atlantic Canada pose to seabirds is linked not only to the severity of storms but also their timing. The current “hurricane season” coincides with part of the breeding phenology of Northern Gannets and Leach’s Storm-Petrels (Fig. 1) such that hatch-year birds are fledging during September and October, when the frequency of tropical storms has been high, historically (Table 1). Though natural weather phenomena have always occurred in areas with high seabird presence, some research suggests that climate change could shift the timing of the “hurricane season” by ~1–2 days earlier each year, with waters warming more rapidly at higher latitudes (Truchelut et al., 2022), and that cyclones could maintain their intensity to higher latitudes. With the frequency and severity of extreme weather events projected to due to climate change (Knutson et al., 2010; Knutson & Tuleya, 2004; United States Environmental Protection Agency, 2016), mortality associated with tropical cyclones is also expected to increase, particularly if stronger storms occur earlier in the year when chicks are too young to survive without parental care and other seabird species are fledging. If tropical cyclones make landfall in Newfoundland earlier in the season, other seabird species breeding in this region may face mortality risk. For example, Atlantic Puffins had the third longest and latest breeding season relative to other colonial K-selected seabirds in Newfoundland (Fig. 1), with chicks fledging throughout August into early September (Runnells, Montevecchi & Davoren, 2024; Wilhelm et al., 2021) and most adults having departed the colony by the end of August (Nettleship, 1970). Razorbills would also be at risk to the effects of tropical cyclones arriving earlier, as fledge dates at most Atlantic colonies appear to be in mid-August (Lavers, Hipfner & Chapdelaine, 2020; Runnells, Montevecchi & Davoren, 2024). Furthermore, the earlier onset of tropical cyclones at high latitudes may impact younger, smaller, and more precocious chicks who cannot go to sea or fly to evade the severe weather conditions, leading to higher mortality across species at breeding colonies. Some evidence indicates that alcids and kittiwakes are trending toward later breeding seasons despite the correlational evidence that breeding phenology advances earlier in the year in response to climate change (Wanless et al., 2009). If this trend holds for alcids and kittiwakes in the northwestern Atlantic, the exposure of the seabirds breeding in Canada to the risk of hurricanes in the region will worsen regardless of a shift in the timing of peak hurricane seasons.

Conclusions

Tropical cyclones negatively affect seabirds in the North Atlantic. The history of tropical cyclones in Newfoundland shows frequent overlap with seabird breeding sites. The peak “hurricane season” in Newfoundland is demonstrated to be from August through October, which leaves seabirds with life histories tied to late fledging periods, such as Leach’s Storm-Petrels and Northern Gannets, most vulnerable to the risks associated with tropical cyclones. As a case study, we explored a Category 1 hurricane in 2021, Hurricane Larry, and found a correlation in the timing of that hurricane with an increase in the number of stranded and dead Northern Gannets and Leach’s Storm-Petrels. Climate change is altering the timing, frequency, severity, and attributes of tropical cyclones, with predictions of stronger, more frequent storms occurring earlier at higher latitudes. This outcome will likely worsen the impacts that tropical cyclones have on seabirds throughout the North Atlantic, particularly at dense breeding sites where they are most vulnerable. Moving forward, it is essential that researchers closely monitor seabirds that face the direct impacts of tropical cyclones and have a protocol in place to ensure a proactive and effective rescue effort for seabirds.

Supplemental Information

Supplemental Information 1 Northern Gannet and Leach’s Storm-Petrel colony information as of 2021.

We thank Chris Mooney, Katharine Studholme, and Bronwyn Harkness for their assistance with data collection. We thank Joshua Mack for his contributions to data collation. Thank you to the Canadian Parks and Wilderness Society—Newfoundland and Labrador Chapter (CPAWS-NL) for providing stranding data for Leach’s Storm-Petrels. We thank Tsz-Kin (Eric) Lai for assistance with reviewing the tropical cyclone information.

Additional Information and Declarations

Competing Interests

The decision to publish the manuscript was part of a contract between Tori V. Burt and Environment and Climate Change Canada.

Author Contributions

Tori V. Burt conceived and designed the experiments, performed the experiments, analyzed the data, prepared figures and/or tables, authored or reviewed drafts of the article, and approved the final draft.

Robert J. Blackmore conceived and designed the experiments, performed the experiments, analyzed the data, prepared figures and/or tables, authored or reviewed drafts of the article, and approved the final draft.

Sydney M. Collins conceived and designed the experiments, performed the experiments, analyzed the data, prepared figures and/or tables, authored or reviewed drafts of the article, and approved the final draft.

Kyle J. N. d’Entremont conceived and designed the experiments, performed the experiments, authored or reviewed drafts of the article, and approved the final draft.

Christopher R. E. Ward performed the experiments, authored or reviewed drafts of the article, and approved the final draft.

Joshua Cunningham performed the experiments, authored or reviewed drafts of the article, and approved the final draft.

Cerren Richards conceived and designed the experiments, performed the experiments, authored or reviewed drafts of the article, and approved the final draft.

Fiona Le Taro conceived and designed the experiments, performed the experiments, analyzed the data, authored or reviewed drafts of the article, and approved the final draft.

Sabina I. Wilhelm performed the experiments, authored or reviewed drafts of the article, and approved the final draft.

Amanda E. Bates conceived and designed the experiments, authored or reviewed drafts of the article, and approved the final draft.

Stephanie Avery-Gomm conceived and designed the experiments, authored or reviewed drafts of the article, and approved the final draft.

William A. Montevecchi conceived and designed the experiments, authored or reviewed drafts of the article, and approved the final draft.

Animal Ethics

The following information was supplied relating to ethical approvals (i.e., approving body and any reference numbers):

Northern Gannets were collected under Canadian Wildlife Service scientific permit no. SC4063 and Leach’s Storm-Petrels were collected under a CPAWS-NL Canadian Wildlife Service permit no. LS2688.

Field Study Permissions

The following information was supplied relating to field study approvals (i.e., approving body and any reference numbers):

Field research was approved by the Canadian Wildlife Service (scientific permit numbers SC4063 and LS2688).

Data Availability

The following information was supplied regarding data availability:

The data and code are available at Figshare:

- Burt, Tori; Blackmore, Robert; Collins, Sydney M.; D’entremont, Kyle J.N.; Ward, Christopher; Cunningham, Joshua T.; et al. (2025). Data and code for “Impacts of Tropical Cyclones on Northwest Atlantic Seabirds: Insights from a Category 1 hurricane”. figshare. Dataset. https://doi.org/10.6084/m9.figshare.28566530.v1.

The wind radii data is available at the National Oceanic and Atmospheric Administration: https://www.nhc.noaa.gov/data/tcr/index.php?season=2021&basin=atl.

The data is available in the Supplemental File.

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
