# Peer review of "Impacts of tropical cyclones on Northwest Atlantic seabirds: insights from a Category 1 hurricane"

_PeerJ, doi:10.7717/peerj.20157_

## Round 0.1 · original submission · Minor Revisions

I’m sorry this is so terse. I’m in the field with no internet. Please respond to these comments and have someone who understands the weather systems look over what you have about them.

·

Basic reporting

Altogether, I thought the authors clearly state their intent, structure the manuscript well, and use the best data available to support their conclusions.

The figures were all appropriate and support their Results. I particularly liked Figure 1, which was very clear and well-designed.

One minor comment: The Introduction, while acceptable, is likely longer than it needs to be. Removing some details on how hurricanes form and general seabird ecology would make for a more concise Introduction.

Experimental design

“Impacts of Tropical Cyclones on Northwest Atlantic seabirds: Insights from a Category 1 hurricane” documents the impact hurricanes have on seabird colonies in Newfoundland. While it primarily focuses on a specific case study of Hurricane Larry in 2021, I think these types of records are essential to publish as part of larger ongoing monitoring efforts.

The Methods were generally clear in describing how the data were collected. They are straightforward and relevant to the question being asked. In other study locations, I'd be interested to see more attention to cumulative wind speed across multiple storms of varying Categories, but Table 1 convincingly shows that is not relevant here.

One minor comment: I found the Leach’s Storm-Petrel Methods section a bit confusing to follow at times since the timeline bounces back and forth a few times. It was not always immediately clear when the authors were referring to 2021 vs other years (such as L247).

Validity of the findings

The manuscript does a convincing job of linking a direct hurricane strike to increased strandings of birds still breeding. The authors have systematically shown that hurricanes overlap spatially and temporally with the breeding of Northern Gannets and Leach's Storm-Petrels in Newfoundland. They then demonstrate that Hurricane Larry led to an unusual number of strandings as compared to other years without a hurricane.

Personally, I found the author's hypothesis that burrow type changes the risk of strandings between the two species to be both interesting and reasonable.

Additional comments

Last minor comment: The authors are correct to note that an increase in hurricane frequency from climate change could be an issue, but it’s not clear how big a concern hurricanes are for this colony. Yes, the authors have convincingly shown that hurricanes increase strandings, but the numbers observed are a tiny fraction of the number of breeding pairs. I recognize that a population viability analysis is likely out of the scope of this paper, but some reference to how concerning this phenomenon is (other than the fact that seabirds are K-selected species) would make this paper more impactful.

Reviewer 2 ·

Basic reporting

The manuscript is clearly written in professional and accessible English, making the reading of it interesting and easy to follow. The structure/ formation of the document follows the PeerJ rules. Background and rationale of the study are well supported by a comprehensive literature review and contextualization of the conducted work.

Experimental design

Research questions are clearly defined, within the scope of PeerJ and directly address a relevant/ pertinent knowledge gap. Methods are generally well described and ethical permits are documented.

Yet, I suggest the authors should more explicitly acknowledge the spatial bias of stranding reports and the limitations of using citizen science as a primary source of mortality data. This should be evident still during the introduction section, not extensively developed, but with few lines referring to this issues.

Validity of the findings

Results reported on this manuscript on the effects of Hurricane Larry provide convincing evidence that tropical cyclones can cause elevated mortality and strandings in seabirds. Results are presented with appropriate caution, i.e. not over interpreting the findings of this study and contextualized with historical comparisons.

Nevertheless, at the discussion section the authors should debate/ interpret how these mortality events compare to background mortality rates or overall colony productivity to assess population-level impacts.

Additional comments

This manuscript makes a valuable contribution to the understanding of how tropical cyclones impact breeding seabirds in the North Atlantic, especially in the context of climate change.
Overall, I commend the authors for assembling a unique and meaningful dataset and for drawing attention to an underreported source of seabird mortality.

Specific suggestion of changes:

L101 - 112: I applaud the authors for clearly describing their study questions, but I urge them to also add specific expected results to those at the end of the introduction. Such expectations would then be confronted with the actual results of their work and on Discussion section, authors would give support to those expectations or advance alternative reasons for different results from the impact of hurricanes on the five seabird species occurring in the area.
The authors should create a new barplot to add to the supplementary material, summarizing observed stranding/mortality events by all other species (besides Leach’s Storm-Petrels) and for all study years to complement Figure 6.

·

Basic reporting

The manuscript is well-written and the figures are useful and aesthetically pleasing. The raw data is provided in organized tables.

Major concern 1: The introduction needs editing by an expert on meteorology -- the sections summarizing tropical cyclone behavior and weather in the region are not entirely correct. See more detailed comments below.

Experimental design

The manuscript addresses a relevant and interesting question with clear methods.

Major concern 2: The work is lacking statistical analysis. I appreciate that the number of years is small and population data collection is difficult, so one has to work with the data one has. But, for example, the number of strandings and deaths for September 10-14, 2021 is said in the abstract to be “significantly” higher than that for the same days in four other years. Significant by what metric? Five years is not enough data to make a stranding climatology for a four day period, so the authors would need to redo this analysis in a more sophisticated way to make a statistically convincing argument. Or they can place these numbers in a larger context of stranding and population data to give a sense of the distributions involved. As written, the paper lacks the statistical context to let the reader assess the magnitude of these strandings and deaths.

Validity of the findings

The results are laid out clearly and the discussion section has interesting and nuanced interpretations, such as the different nest types affecting the resiliency to tropical cyclones.

Major concern 3: I am confused by the framing of the results, which seems ambiguous. Do tropical cyclones have a major negative effect on these bird populations or not? The abstract was written in a way that suggested large stranding/death events from Hurricane Larry and other tropical cyclones. But the results didn’t seem to support this, and by the time I got to the discussion section, the authors said, “Compared to other stranding and mortality events associated with tropical cyclones, the impacts of Hurricane Larry were minimal.” But then the conclusion opens with, “Tropical cyclones negatively affect seabirds in the North Atlantic.” So what is the message of the paper? Is Hurricane Larry representative of hurricane effects in this region? The manuscript seems to be stuck in between whether or not these tropical cyclones are important factors in the seabird population dynamics. I would suggest picking a central message of your results and using it as a through-line from abstract to conclusion. If I were a seabird conservation planner, what actionable insights should I be taking away from this?

Additional comments

Detailed feedback:

Abstract: “Raise the spectre” is colloquial and vague. For the abstract, I suggest giving a more definitive statement of the expected impacts of climate change.

47: "Ocean climates" is a confusing term. I suggest referring to “global climate” instead.

50: Saffir-Simpson is for Atlantic and Eastern Pacific TCs only. I suggest a slight clarification there.

63: What is the relevance of disease here? Is it tied to TCs at all?

78: “Spatiotemporally interact” is an awkward way to phrase this

82-87: This summary of TC behavior is not entirely correct. Here are a few notes:

1. It is an oversimplification to say that they track northward due to pressure gradients. I would instead say that their track is controlled by the steering flow from the subtropical high, extratropical westerlies, and low pressure systems.
2. They sometimes strengthen when moving northward, but often they do not. The strengthening at subtropical latitudes is due to the high SSTs rather than pressure gradients. If the TC survives farther northward, it can undergo extratropical transition, which would be relevant for many storms that reach NE Canada. If it doesn’t undergo extratropical transition, the hurricane will die out due to lack of an SST energy source once it reaches cold northern waters.
3. The strongest winds in a TC are in the front-right quadrant relative to the storm motion due to the combination of the rotational wind speed with the storm forward movement. I would state this more precisely in the manuscript.

118-121: This description of weather factors isn’t totally correct either:

1. The jet stream doesn’t “end” anywhere; it’s circumglobal. I would just say that Newfoundland is in the region of the polar jet stream.
2. Newfoundland is too far north to experience the trade winds, which are a tropical phenomenon.

135: What is the justification for limiting to Cat 1 hurricanes? These categories are based only on maximum sustained wind speed, which doesn’t necessarily determine the storm impacts. For instance, a tropical storm with a very large wind field and pouring rain can have stronger impacts than a compact Cat 1 hurricane.

162: Witless Bay listed twice

281-282: “At least one has passed over all six” — confusing phrasing. Reword to make it more clear what this means.

328-331: 35 individuals doesn’t sound like a very high number compared to the number of breeding pairs in the table. So was this event consequential? I see now that you address this in the discussion; perhaps mention it with the results too.

405-406: If Larry didn’t have a large impact, why choose it as a case study? Is it because it is the only storm for which you have sufficient data? Or because you want to show that TCs in this region don’t have a large effect on these seabird populations? It would be helpful if you clarified that case study choice in the text.

441: You say that TCs negatively affect seabirds in the North Atlantic, but they have been coexisting for millions of years. I think it would be useful to address this longer perspective in the discussion.

Fig. 4: The circle sizes are hard to tell apart. Can you mention the total number of strandings/deaths in the caption to give a sense of what those circles add up to? You could also mention that they are in the front-right quadrant of the storm.

Fig. 5: Same comment as for Fig. 4.

Fig. 6: Caption says September 10-14, 2020, but the chart shows many years. Does it instead mean the strandings during Sep 10-14 each year? Also, this is not enough years to get good statistics for a specific 4-day period. What if you compare the max number of strandings/deaths over a 4-day period each year instead of looking at those specific days? That would be more statistically robust and give a wider context.

---

## Round 0.2 · accepted · Accept

Thank you for addressing the remaining issues on your manuscript. I am happy that you have dealt with all the reviewers' concerns. Thank you for sending this paper to us for publication.